# MSfusion: Enabling Collaborative Training of Large Models over Resource-Constraint Participants

## Abstract

Training large models like GPT-3 requires a large amount of data, as well as abundant computation resources. While collaborative learning (e.g., federated learning) provides a promising paradigm to harness collective data from many participants, performing training for large models remains a major challenge for participants with limited resources. We introduce MSfusion, an effective and efficient collaborative learning framework, tailored for training large models on resource-constraint devices through *model splitting*. Specifically, a double shifting model splitting scheme is designed such that in each training round, each participant is assigned a subset of model parameters to train over local data, and aggregates with sub-models of other peers on common parameters. While model splitting significantly reduces the computation and communication costs of individual participants, additional novel designs on adaptive model overlapping and contrastive loss functions help MSfusion to maintain training effectiveness, against model shift across participants. Extensive experiments on image and NLP datasets illustrate significant advantages of MSfusion in performance and efficiency for training large models, and its strong scalability: computation cost of each participant reduces significantly as the number of participants increases.

## 1 Introduction

The contemporary technological landscape, characterized by the emergence and rapid evolution of large models, has ushered in a transformative era for machine learning and artificial intelligence. Large language models (LLMs), exemplified by GPT-3 and its counterparts, trained on vast corpora of billions of tokens, have captivated the global community with their remarkable capabilities, including human-like text generation, language translation, question answering, and document summarization Radford et al. (2018); Zhao et al. (2023). However, the practical training of these models is hamstrung by substantial computational and data requirements.

Consider the following real-world scenario: multiple companies, each armed with its own resource-limited servers (or cloud instances) and the private data collected from their respective clients, aspire to harness the advantages of large models. The objective, therefore, is to leverage the existing computational power of their servers collaboratively to train a high-performance large model. Additionally, due to privacy and cost considerations, the introduction of an additional central server is unsuitable for these companies. Conventional distributed learning methods like FedAVG McMahan et al. (2017) is not applicable as it is not practical to perform local SGD on large models, given the memory and computation constraints on companies' local servers. As demonstrated in Dey et al. (2023), utilizing only $10\%$ of a large language model during training can result in up to a 100-fold reduction in Floating Point Operations (FLOPs), translating to substantial cost savings. Motivated by this, we ask the following question: *Is it possible for these companies to collaboratively train a high-performance large model over their private data, with each company training a sub-model as a split from the full model?*

To address the above question, we propose MSfusion, a novel collaborative learning framework that utilizes model splitting to enable effective and efficient training of large models over resource-constraint participants. MSfusion leverages a network of decentralized participants, each equipped

with its unique dataset, to independently extract and train split models from a larger model, effectively managing resource constraints. A novel double shifting splitting scheme is proposed to ensure extensive coverage of the global full model by the participants. An overlap aggregation method is introduced to further reduce communication needs. Moreover, an adaptive splitting mechanism is introduced to dynamically adjust the overlap of model parameters across participants as training progresses, expediting model convergence. A contrastive objective is designed to mitigate model drift caused by heterogeneous data distributions and differences in participants' sub-models.

`MSfusion`, as a combination of model and data parallelism, not only reduces computation and communication costs, but also enhances the model performance via utilizing diverse datasets across multiple participants. We implement `MSfusion` and evaluate it over various image and NLP datasets. Extensive experiments demonstrate the substantial advantages of `MSfusion` in model performance and computation and communication efficiencies, over SOTA distributed learning methods using model splitting. `MSfusion` also exhibits strong scalability such that to achieve some target accuracy, the required split model size (hence computation/communication load) of each participant decreases significantly as the number participants increases. We view this as a key enabler for more resource-constraint participant to contribute to and benefit from training of large models.

## 2 RELATED WORKS

### 2.1 DECENTRALIZED LEARNING

Decentralized learning, in contrast to its centralized counterpart, pursues a consensus model through peer-to-peer communication, eliminating the reliance on a central server. This approach offers distinct advantages in terms of communication efficiency and data privacy preservation when compared to Centralized Federated Learning (CFL) Shi et al. (2023); Li et al. (2022). In an exemplary serverless, peer-to-peer Federated Learning (FL) implementation, Roy et al. (2019) introduced BrainTorrent, which has found application in dynamic peer-to-peer FL environments, particularly in medical contexts. Dai et al. (2022) proposed a novel approach employing personalized sparse masks to train personalized models, reducing communication costs by filtering out parameter weights with minimal influence on the gradient.

### 2.2 KD-BASED METHOD

Knowledge Distillation (KD) based methods offer a solution wherein the complex knowledge encapsulated in a large model (server model) is imparted to a smaller, more tractable model (client model). Methods such as FedET in Cho et al. (2022) have demonstrated some efficiency of KD. The defining strength of KD-based methods lies in their ability to train more compact models, approximating the performance of their larger counterparts with substantially less computational overhead. Nonetheless, achieving competitive accuracy typically necessitates access to public datasets that align in domain and scope with the client data Lin et al. (2020). A critical constraint of KD-based methods is the necessity for a central server, both for computational intensity and compatibility with decentralized architectures and secure aggregation protocols, rendering them less suited to the decentralized collaborative settings introduced in this study.

### 2.3 MODEL PARTIAL TRAINING

The partial training (PT-based) paradigm presents a distributive strategy wherein the model is fragmented across multiple servers, and each server is responsible for training a discrete segment of the model Hong et al. (2022); Alam et al. (2023); Diao et al. (2021). This modular approach considerably alleviates the computational demands on individual servers and fosters parallelized training. A notable limitation of current PT-based methodologies is their confinement to CFL frameworks, designed to reconcile computational disparities among clients. Such methods inherently assume the participation of participant possessing the complete model during the training phase, a presumption misaligned with the decentralized scenario envisaged in our work. And these participant actually play a important role, without such participants these methods failed to obtain a good performance shown in Section 5. Shulgin & Richtárik (2023) provide more detailed theoretical understanding behind PT-based methods, further showing its potential.

## 3 PROBLEM DEFINITION

Consider a collaborative learning system of $N$ participants. Each participant $n$, $n \in \{1, ..., N\}$, has a local dataset $\mathcal{D}_n = \{(x_i^{(n)}, y_i^{(n)})\}_{i=1}^{M_n}$ with $M_n$ collected samples. The goal is to train a global model $W$ over all participants' datasets, to solve the following optimization problem:

$$\min_{W} \ f(W) = \frac{1}{N} \sum_{n=1}^{N} \hat{f}_n(W),$$

$$\text{s.t.} \ \hat{f}_n(W) := \frac{1}{M_n} \sum_{i=1}^{M_n} \hat{\mathcal{L}}(W; (x_i^{(n)}, y_i^{(n)})). \tag{1}$$

Here $\hat{f}_n(W)$ is the local empirical risk of participant $S_i$, for some loss function $\hat{\mathcal{L}}$.

We focus on the scenarios of collaborative training of large models (e.g., LLMs with billions of parameters), where it is impractical for a participant to locally train $W$ due to limited memory and computation resources. We consider a model splitting framework, such that each participant trains a smaller sub-model $w_n$ split from the global full model $W$ ($w_n \subseteq W$). Based on this, the local empirical risk for participant $n$ becomes

$$f_n(w_n) := \frac{1}{M_n} \sum_{i=1}^{M_n} \mathcal{L}_n(w_n; (x_i^{(n)}, y_i^{(n)})), \tag{2}$$

where $\mathcal{L}_n$ is the local loss corresponding to the sub-model $w_n$. After the participants finish their local training, the obtained sub-models are fused into a global model. This model fusion can take place over many rounds, and the sub-model trained at each participant can vary across rounds.

We define *split model size* of participant $n$, denoted by $\mu_n$, as the ratio of the size of $w_n$ to the size of $W$, i.e., $\mu_n = \frac{|w_n|}{|W|}$. In practice, $\mu_n$ is principally determined by the computation and communication capabilities of the participant. While previous studies have often assumed the presence of a powerful participant who can process the entire model, i.e., $\mu_n = 1$, we focus primarily on the scenario where a group of less capable participants collaborate to train an effective large model, where all participants have comparable but small split model sizes, e.g., $\mu_n \leq 0.5$ for all $n$.

Training large models over the collaborative learning framework described above are faced with following major challenges.

- **Efficiency:** Computation cost for local training and communication cost to exchange models/gradients are major efficiency bottlenecks when dealing with large models Chen et al. (2022); Zhang et al. (2023). Although model splitting helps to alleviate this issue, doing it naively may significantly reduce the model performance.

- **Data and Model Heterogeneity:** Like in the case of FL, the local data on different participants tend to follow different distributions; in addition, the split portions of the global full model may diverge among participants. As it is well known that data non-iidness leads to reduced model performance Collins et al. (2021), the "model drift" caused by double heterogeneity of sub-model and local data poses serious challenges on training effective large models.

- **Scalability:** To encourage participation of more resource-limited devices, it is desirable that as the number of participants increases, a smaller split model size is required on each participant to achieve a target accuracy. However, more participants exacerbates the issue of model drift, potentially degrading the model performance. How to design model splitting to maintain model performance with reduced split model size is hence crucial to achieving scalable collaborative training.

## 4 MSFUSION

In this section, we introduce `MSfusion`, a model splitting approach to address the above challenges, for effective and efficient collaborative training of large models. Figure 1 provides an overview of proposed `MSfusion`. Algorithm 1 gives the pseudo-code of `MSfusion`.

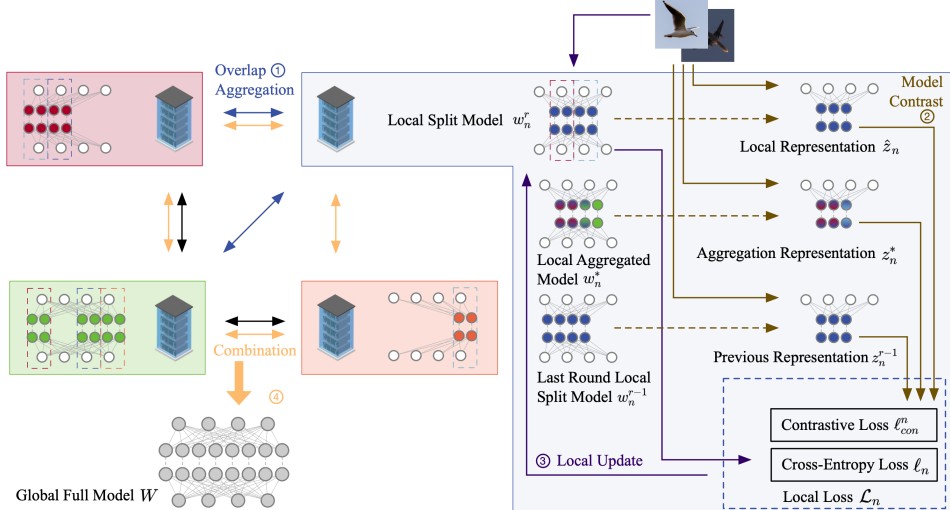

Figure 1: Overview of MSfusion.

## 4.1 DOUBLE SHIFTING SPLITTING SCHEME

We first introduce a novel splitting approach, dubbed Double Shifting Model Splitting Scheme (DSS), as essential part of our `MSfusion` framework for achieving model partitioning among participants. `MSfusion` employs a two-tier shifting model partitioning approach: one at the inter-participant level and another at the inter-round level. This departs significantly from existing methodologies such as Federated Dropout, which utilizes random splitting; HeteroFL and FjORD, which employ static splitting; and FedRolex, which implements round-rolling splitting. Figure 2 shows the difference between DSS and previous model splitting schemes.

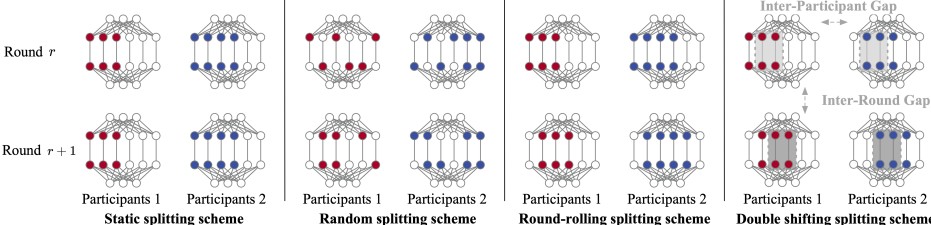

Figure 2: Difference between DSS and previous model splitting schemes.

**Inter-Participant Gap:** During either the initiation phase of training or when there is a change in the set of participants (either through joining or leaving) each participant is assigned an unique index through consensus. For participants with adjacent indices, the starting nodes of their sub-models at $i$-th hidden layer of the global full model is differed by the following gap

$$G_i = \frac{K_i}{N} \times c, \tag{3}$$

where $K_i$ represents the size of $i$-th hidden layer, and $c \in [0, 1]$ serves as an overlapping control parameter. Consequently, the index of the starting node at participant $n$ is $nG_i$. This gap ensures that the entirety of the global model $W$ is adequately represented across all participants during a single communication round. Moreover, it ensures overlap (shade part in Figure 2) between sub-models assigned to adjacent participants.

**Inter-Round Gap:** Between successive communication rounds, each participant shifts the starting node of its local sub-model by a gap $\zeta$. This gap ensures that the parameters of the global model $W$ are uniformly optimized by individual participants. In `MSfusion`, $\zeta$ is set to 1.

For a participant with index $n$ and a split model size $\mu_n$, for the $i$-th hidden layer of the global full model, the indices of the neurons (for fully-connected layers and hidden layers of attention heads of transformers) or filters (for convolutional layers) contained in the sub-model $w_n$ in round $r$, denoted

by $\varphi_{n,i}^{(r)}$, is presented as follows.

$$\varphi_{n,i}^{(r)}=\begin{cases} \{\mathcal{G}_{n,i,r}, \mathcal{G}_{n,i,r}+1, \ldots, \mathcal{G}_{n,i,r}+\lfloor\mu_n K_i\rfloor-1\} & \text{if } \mathcal{G}_{n,i,r}+\lfloor\mu_n K_i\rfloor \leq K_i, \\ \{\mathcal{G}_{n,i,r}, \mathcal{G}_{n,i,r}+1, \ldots, K_i-1\} \cup \{0, 1, \ldots, \mathcal{G}_{n,i,r}+\lfloor\mu_n K_i\rfloor-1-K_i\} & \text{else.} \end{cases}$$
(4)

Here $\mathcal{G}_{n,i,r} = nG_{n,i} + r\zeta$ is the combined Inter-Participants and Inter-Round Gap. Note that in DSS the split model size $\mu$ dictates the number of output channels/neurons for the corresponding layer, which subsequently influences the number of input channels/neurons for the succeeding layer. To preserve dimensional consistency, DSS is intentionally not applied to the input and final output layers of the model. **Detail analysis for DSS are provided in Appendix A.1.**

## 4.2 OVERLAP AGGREGATION

The introduction of `MSfusion` brought with it a transformative change in model splitting by leveraging the DSS. A notable innovation is the implementation of a stable, controllable overlap, expressed as $\Psi_{\{n,n+1\},i}^{(r)} = w_{n,i}^{(r)} \cap w_{n+1,i}^{(r)}$. Such an overlap encapsulates the intersection of adjacent participant models, distinctly differentiating `MSfusion` from prior methodologies.

Traditional methods, such as static and round-rolling splits, consistently retain overlaps equal to the smaller-sized participant models, so that in each communication round only this small part of global model $W$ is collaboratively aggregated. While the random splitting method yields overlaps that are unstable for participants to utilize. `MSfusion`, however, simplifies overlap recognition between participants and provides robust control with the overlapping control parameter, $c$. This overlap rate can be precisely quantified as: $\delta_{\{n,n+1\}} = 1 - \frac{c}{\mu N}(c < \mu N)$ for two adjacent participant with the same $\mu$. And for random two participant $\delta \in [0, 1 - \frac{c}{\mu N}](c < \mu N)$.

**Overlap Aggregation:** Differencet from FedAvg McMahan et al. (2017), where server aggregation rely on the complete models submitted by clients. `MSfusion` mandates only the overlapping segments for transmission among connected participants. That is for each two connected participant only send and receive the overlapping part between them. And this can be formalized as the following overlapping average in `MSfusion`:

$$\theta_{n,[l,i]}^* = \frac{1}{S+1} \sum_{s_i \in S} (\theta_{s_i,[l,i]}^r + \theta_{n,[l,i]}^r)$$
(5)

where $\theta_{n,[l,i]}^r$ donates the $i-$th parameter of layer $l$ of server $n$ from $w_n^r$, $S \in N$ is the connected participants holding $\theta_{n,[l,i]}^r$. This tailored aggregation around overlaps considerably diminishes communication overheads, offering an efficient conduit for training LLMs in a distributed manner.

`MSfusion`'s initial setup hinges on a consensus regarding the participant index and $\mu_n$, ensuring ease in determining overlaps. During subsequent training iterations, each participant only train its split model obtained from DSS, the global full model is not stored. Instead, an efficient fusion mechanism within `MSfusion` fetches the requisite global model. Participants engage with adjacent peers to access missing parameters ($\complement_{K_i}\Theta_{n,i}^{(r)}$), and combing these with aggregated overlap parameters thus effectively obtaining the global full model $W$ for further inference.

**Dynamic Overlap to Boost Convergence:** We design adaptive overlap strategy to further speed up the convergence of `MSfusion`. In each 10 rounds($r \mod 10 = 0$), `MSfusion` updating the $c$ according to

$$c = c_0(1 - (r/R)p^*)$$
(6)

Where $c_0$ is the given initial control parameter, $p$ is the final stage parameter, $R$ is the total round number. Since there is inter-round gap $c$ is not updated per-round. At the training onset, a constant value of $c_0 = 1$ is chosen to yield a smaller overlap rate $\delta$, enabling participants to encompass more of the global full model, thereby accelerating training. As training progresses, $c$ is tapered to amplify $\delta$, emphasizing collaborative fine-tuning. It's imperative to underscore that a larger $\delta$ isn't always advantageous, given its direct implications on communication overhead and global model coverage.

## 4.3 CONTRASTIVE OBJECTIVE

In the considered collaborative learning framework, data on different participants often have distinct distributions, leaning to model shift after local training. In addition, model splitting across partici-

---

**Algorithm 1:** `MSfusion`

---

**Input:** Multi-server set $N_n$, model split rate $\mu_n$, local dataset $\mathcal{D}_n$, initial global full model $W^0$, final stage parameter $p$.

**Output:** Server maintained split model $w_n^*$, trained the global full model $W^*$.

**Initiation:**

Assign an index to each participant based on network topology by consensus.

**Per-Server Operations:**

**while** round $r < R$ **do**

    **if** $r \mod 10 = 0$ **then**

        Update $c = c_0(1 - (r/R)p)$ in $\mathcal{G}_{n,i,r}$;

    Split local $w_n^r$ from $W^r$ by DSS (4) with $\mathcal{G}_{n,i,r}$;

    Transmit $\theta_{n,[l,i]}^r$ to all connected servers holding $i-$th parameter;

    Receive $\theta_{s_i,[l,i]}^r$ from $S$ connected servers holding $i-$th parameter;

    Aggregate local parameters based on (5)

    Update $w_n^*$ based on $\theta_{n,[l,i]}^*$, and representations $z \leftarrow w_n^r$, $z^* \leftarrow w_n^*$, $z^{r-1} \leftarrow w_n^{r-1}$.

    Sample batch $b = \{(x_i, y_i)\}_{i=1}^B$ from $\mathcal{D}_n$;

    The combined loss: $\mathcal{L}_n = \ell_n + \lambda \ell_{con}^n$, the contrastive loss (7)

    Update: $w_n^{r+1} \leftarrow w_n^r - \eta \nabla \mathcal{L}_n$

    Receive round gap model parameter $\theta_{s,[l,i]}^{r+1}$ from connected server.

**Global Model Combination:**

Transmit $\theta_{n,[l,i]}^R \in w_n^R$ to connected server lacking $i-$th parameter;

Receive $\theta_{s,[l,j]}^R \in \complement_W w_n^R$ from connected server holding $j-$th parameter;

Combine $\theta_{n,[l,i]}^R$ with $\theta_{s,[l,j]}^R$ to obtain the global combined model $W^*$.

---

pants further exacerbate this drift, potentially undermining model performance. In the `MSfusion` framework, the local aggregated model at each participant can be viewed as a potent surrogate for the global full model. This feature inherently lends itself to the adoption of a contrastive learning strategy Chen et al. (2020); He et al. (2020). Nevertheless, a direct application of contrastive FL methods from systems like MOON Li et al. (2021) or CreamFL Yu et al. (2023) is untenable, since it is computationally infeasible to run global-local model contrast at the scale of full model size.

In `MSfusion`, we apply contrastive learning on sub-models to curb the divergence between a participant's local model and the corresponding aggregated model. For any input $x$, `MSfusion` extracts its local representation $\hat{z}_n$ from the current sub-model $w_n^r$ that being updated, the aggregation representation $z_n^*$ from the local aggregated model $w_n^*$, and the previous representation $z_n^{r-1}$ from the sub-model of last round $w_n^{r-1}$. Note there exist a inter-round gap shift between sub-models in consecutive rounds. We focus on representations on the common part between $w_n^r$ and $w_n^{r-1}$. We construct the contrastive loss in `MSfusion` as follows:

$$\ell_{con}^n = -\log \frac{\exp\{[(\hat{z}_n)^T \cdot z_n^*]/\tau\}}{\exp\{[(\hat{z}_n)^T \cdot z_n^*]/\tau\} + \exp\{[(\hat{z}_n)^T \cdot z_n^{r-1}]/\tau\}}, \tag{7}$$

where $\tau$ is the temperature coefficient to control the penalties on hard negative pairs Wang & Liu (2021). This loss metric is optimized to align the local split model's representation closely with that of the local aggregated model, thereby curtailing participant model drift.

The overall objective for any participant $n$ when given an input $(x, y)$ is framed as:

$$\mathcal{L}_n = \ell_n(w_n^r; x, y) + \lambda \ell_{con}^n. \tag{8}$$

Here $\ell_n$ is cross-entropy loss, $\lambda$ is the coefficient governing the weight of the contrastive loss.

## 5 EXPERIMENTS

In this section, we present a comprehensive evaluation of `MSfusion` on various benchmark datasets, spanning both image classification and natural language processing tasks. For more experiments details (parameter size, data partition, and model architecture) and results are given in Appendix A.2.

**Datasets** We conduct our evaluations on two distinct task categories: image classification and natural language processing. For image classification, our method is subjected to rigorous testing on three widely recognized datasets: CIFAR10, CIFAR100, and TinyImageNet. For natural language processing (NLP) tasks, our evaluation leverages the PennTreebank, WikiText2 and WikiText103 dataset. To gauge performance on NLP tasks, we employ the perplexity metric, with lower values signifying better performance.

**Models** To underscore the versatility of our approach across varied architectures, we employ both convolutional and transformer models in our experiments. In image classification tasks, a modified ResNet18 following Diao et al. (2021) is employed. Furthermore, the transformative potential of MSfusion is exemplified through its adeptness in handling LLMs. To this end, transformer models are the backbone of our NLP tasks. The global full model parameter size for transformers applied to NLP tasks is 7.32M for PennTreebank, 19.3M for WikiText2 and 139.01M for WikiText103.

**Data Heterogeneity** In the image classification tasks, we introduce data heterogeneity by deliberately skewing the label distribution among participants. This non-iid characteristic is attained by assigning each participant a distinct subset of $H$ classes. Specifically, for CIFAR10, we set $H = 5$, for CIFAR100, $H = 20$, and for TinyImageNet, $H = 40$. Moreover, for WikiText2, we naturally generate non-IID data distribution through dataset partitioning among participants. Additionally, we reduce the vocabulary size from 33,728 to approximately 3,000 words for each participant.

**Spilt model size** Throughout our study, we predominantly focus on participants with either **uniform** small model splits $\mu_n \in \{6.25\%, 10\%, ..., 62.5\%\}$. The global full model represents an unsplit, complete model. **There is no participant with $\mu_n = 1$.** This is a notable advantage compared to previous works with heterogeneous settings. While a participant training the global full model plays an important role in their experiments, it is not present in our case due to large model size. To create local sub-models, we adjust the number of kernels in convolution layers for ResNet18 while keeping the output layer nodes constant. For Transformer models, we vary the number of nodes in the hidden layers of the attention heads.

**Baselines** We compare MSfusion against SOTA PT-based model-heterogeneous FL methods including HeteroFLDiao et al. (2021) and FedRolexAlam et al. (2023), as well as SOTA KD-based FL method Fed-ETCho et al. (2022). To ensure equitable comparisons, we maintain uniformity in parameters across all PT-based baselines, including learning rate, and the number of communication rounds. For ring topology tests, we compare MSfusion with D-PSGD Lian et al. (2017) and the state-of-the-art Dis-PFL Dai et al. (2022).

Table 1: Global model accuracy and computation cost comparison. For NLP tasks, since Fed-ET cannot be directly used for language modeling tasks, result is marked as N/A. FLOPs denotes the floating operations for each participant per round. 10 participants with 100% selected rate.

| | CIFAR10 | | | | CIFAR100 | | | | TinyImageNet | | | |
|---|---|---|---|---|---|---|---|---|---|---|---|---|
| Methods | iid | non-iid | FLOPs | $\mu_n$ | iid | non-iid | FLOPs | $\mu_n$ | iid | non-iid | FLOPs | $\mu_n$ |
| HeteroFL | 40.50 ± 1.2 | 37.36 ± 0.6 | 35.76M | 25% | 19.49 ± 0.9 | 12.11 ± 0.7 | 35.76M | 25% | 11.08 ± 0.6 | 14.54 ± 0.4 | 1.75B | 62.5% |
| FedRolex | 76.98 ± 0.7 | 66.41 ± 0.8 | 35.76M | 25% | 42.33 ± 0.8 | 36.61 ± 1.0 | 35.76M | 25% | 37.05 ± 1.1 | 20.63 ± 0.8 | 1.75B | 62.5% |
| Fed-ET | 83.42 ± 0.3 | 81.13 ± 0.3 | 1.09B | N/A | 41.61 ± 0.4 | 35.78 ± 0.5 | 1.09B | N/A | 29.61 ± 0.4 | 19.78 ± 0.6 | 6.12B | N/A |
| MSfusion S | 78.74 ± 0.6 | 71.21 ± 0.4 | **6.95M** | 10% | 43.77 ± 0.5 | 37.01 ± 0.7 | **6.95M** | 10% | 12.62 ± 0.7 | 11.45 ± 0.5 | **79.79M** | 12.5% |
| MSfusion M | 83.04 ± 0.3 | 75.71 ± 0.4 | 22.33M | 18.75% | 50.04 ± 0.5 | 44.11 ± 0.4 | 22.33M | 18.75% | 39.61 ± 0.6 | 20.91 ± 0.6 | 297.0M | 25% |
| MSfusion L | **87.37 ± 0.5** | **81.91 ± 0.3** | 151.7M | 50% | **60.63 ± 0.4** | **47.21 ± 0.5** | 151.7M | 50% | **51.41 ± 0.4** | **24.67 ± 0.3** | 1.255B | 50% |

| | PennTreebank | | | WikiText2 | | | WikiText103 | | |
|---|---|---|---|---|---|---|---|---|---|
| Methods | Perplexity | FLOPs | $\mu_n$ | Perplexity | FLOPs | $\mu_n$ | Perplexity | FLOPs | $\mu_n$ |
| HeteroFL | 55.97 ± 5.4 | 148.6M | 75% | 579.05 ± 8.4 | 1.08B | 75% | 784.21 ± 23 | 111.9B | 75% |
| FedRolex | 61.52 ± 6.8 | 148.6M | 75% | 547.32 ± 45 | 1.08B | 75% | 697.42 ± 33 | 111.9B | 75% |
| Fed-ET | N/A | | | N/A | | | N/A | | |
| MSfusion S | 9.09 ± 0.7 | **36.2M** | 21.875% | 44.33 ± 3.6 | **198.5M** | 18.75% | 13.21 ± 1.1 | **32.6B** | 21.875% |
| MSfusion M | 8.02 ± 0.5 | 41.8M | 25% | 5.28 ± 0.4 | 290.1M | 25% | 8.92 ± 0.6 | 37.3B | 25% |
| MSfusion L | **3.11 ± 0.2** | 91.4M | 50% | **3.59 ± 0.2** | 633.2M | 50% | **6.31 ± 0.4** | 74.4B | 50% |

## 5.1 PERFORMANCE

Performance and computational cost comparisons between MSfusion and the aforementioned baselines are summarized in Table. 1. Performance metrics are evaluated by testing this unsplit global model on the testing dataset. To ensure equitable comparisons with centralized methods, we apply fully connected topology for MSfusion in these experiments. For Fed-ET, we also calculate the computation cost of the server and average it across participants.

Table. 1 reveals that, particularly for CIFAR100 and CIFAR10 datasets, MSfusion S outperforms HeteroFL and FedRolex with only 10% of the full model per participant, which is less than 1/5 of the computational cost. Increasing the local split model size directly enhances overall performance,

though it comes at the cost of increased computation. `MSfusion` M achieves competitive performance compared to SOTA KD-based method and outperforms all baselines in the more complex TinyImageNet dataset. `MSFusion` L outperforms Fed-ET in all datasets while costing only 1/5 of the computation. Additionally, it's important to note that `MSfusion` does not require a central server and does not rely on public data, in contrast to KD-based methods. In NLP tasks, `MSfusion` significantly outperforms HeteroFL and FedRolex. The suboptimal performance of HeteroFL and FedRolex in collaborative training larger models is due to untrained global model parameters and exacerbated model drift, issues unaddressed in their splitting-focused methodologies. This emphasizes the advantage of `MSfusion` in handling larger models more effectively.

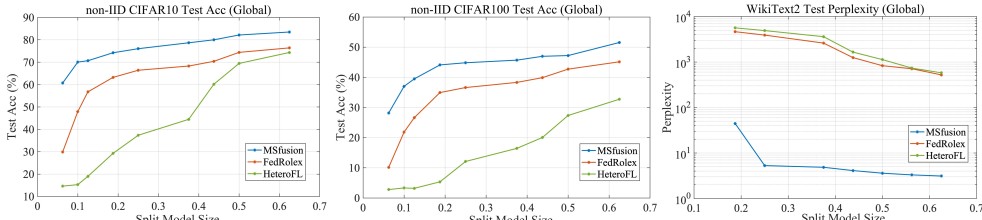

Figure 3: Performance comparisons on non-IID CIFAR10, CIFAR100 and WikiText2 datasets.

Detailed mode size perforamnce comparsion for CIFAR10, CIFAR100 and WikiText2 are shown in Figure. 3. The perofrmance advantage of `MSfusion` is obvious espcially for the smaller split model size to the more efficient DSS scheme. To achieve a target accuracy, much less split model size is required for `MSfusion`, means much less computation and communication cost is required for all participants. Like for CIFAR10, for a target 70% accuracy performance, FedRolex require about 45% split model size which cost 116M FLOPs, while `MSfusion` only require 12.5% split model size which only cost 9.83M FLOPs. That is more that 10 times more computation power for each participants to training a same size global model with ruffly the same performance.

An ablation study on CIFAR10 and Wiki-Text2 with $\mu = 18.75\%$ is given in Table.2. It shows the accuracy and computation comparison for `MSfusion`, `MSfusion` without contrastive objective (w/o Con), `MSfusion` without dynamic overlap (w/o Dyn), and `MSfusion` without both (w/o Con & Dyn).

| Methods | CIFAR10 ACC | FLOPs | WikiText2 PPL | FLOPs |
|---|---|---|---|---|
| HeteroFL | $37.36 \pm 0.6$ | 35.76M | $579.05 \pm 8.4$ | 1.08B |
| FedRolex | $66.41 \pm 0.8$ | 35.76M | $547.32 \pm 4.5$ | 1.08B |
| MSfusion w/o Con | $73.16 \pm 0.5$ | 20.32M | $9.57 \pm 2.1$ | 257.2M |
| MSfusion w/o Dyn | $73.31 \pm 0.5$ | 22.33M | $7.35 \pm 1.5$ | 290.1M |
| MSfusion w/o Con & Dyn | $70.17 \pm 0.8$ | **20.32M** | $11.24 \pm 2.3$ | **257.2M** |
| MSfusion | **$75.71 \pm 0.5$** | 22.33M | **$5.276 \pm 0.4$** | 290.1M |

Table 2: Ablation studies.

The results showcase `MSfusion`'s superior performance over all baseline variants. Specifically, in NLP tasks, the performance gain of `MSfusion` is more significant. This is mainly attributed to the larger relative size of local model parameters within the transformer architecture, further amplifying the model drift problem. This demonstrate that the proposed contrastive objective and dynamic overlap techniques play key role in ensuring the effectiveness of `MSfusion` training, with a marginal increase in computational cost. Moreover, `MSfusion` without contrastive objective can still outperform all PT-based methods with much less computation cost.

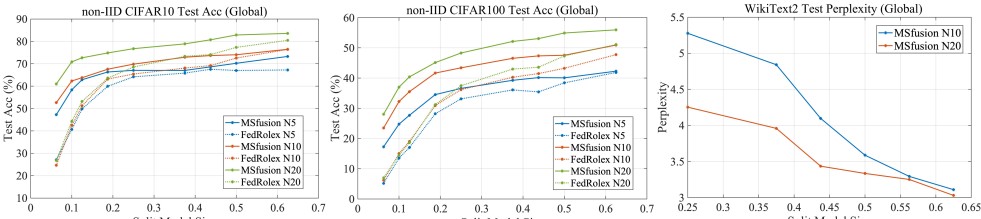

Figure 4: Performance of `MSfusion` for different numbers of participants.

## 5.2 SCALABILITY

In order to assess scalability, we conducted experiments where we held the training data size constant for participants, each assigned a fixed 1/20 portion of the training data (while the test data for the global full model remained unchanged). We then evaluated the performance of `MSfusion` and FedRolex with varying numbers of participants, all employing the same split model size for collaborative training of a larger global full model. The results, presented in Figure. 4, demonstrate that `MSfusion` consistently outperforms the SOTA FedRolex across various participant counts.

Notably, FedRolex faces challenges in ensuring scalability, particularly when dealing with small split model sizes, as it does not adequately address the issue of model drift. To achieve a target level of performance, as number of participants increases, the required split model size on each participant reduces significantly. For instance for CIFAR10, to achieve 70% accuracy, the split model size reduces from 50% with 5 participants to merely 10% with 20 participants. This makes `MSfusion` a key enabler for resource-limited devices to contribute to and benefit from collaborative training.

### 5.3 TOPOLOGY

In scenarios with limited network resources, the utilization of a ring topology becomes a viable option. Figure. 5 (a) illustrates the distinctions between the ring topology and the fully-connected network topology. To utilize `MSfusion` in ring topology, the participants with nearest network topologies, the indexing is designed to be proximate, thereby facilitating optimal communication. We examine the performance of `MSfusion` under these two different network topologies, as detailed in Figure. 5 (b). It is worth noting that the Global Model Combination step in `MSfusion` occurs every 10 communication rounds, primarily for evaluating the performance of the global full model on the testing dataset (this is exclusively for testing purposes and is not required during actual training). Our findings reveal that `MSfusion` effectively facilitates a scalable collaborative training process while maintaining a high level of performance under ring topology.

Since in both Dis-PFL and D-PSGD, each participant retains a local model, and accuracy is calculated as the average of local model accuracy over their respective local datasets. To ensure a fair comparison, `MSfusion` leverages local maintained global full model accuracy over the local dataset, which is then averaged across all participants to determine local accuracy. The results, depicted in Figure. 5 (c), shown `MSfusion` is able to

| Methods | ACC | COMM | FLOPs |
|---|---|---|---|
| Dis-PFL | $52.57 \pm 0.3$ | 44.8MB | 700.1M |
| D-PSGD | $13.38 \pm 0.5$ | 89.7MB | 830.3M |
| MSfusion(ini) | $\mathbf{56.57 \pm 0.6}$ | **2.48MB** | **22.33M** |
| MSfusion(ter) | $\mathbf{56.57 \pm 0.6}$ | **3.21MB** | **22.33M** |

Table 3: Local efficiency comparison for ring topology.

converge significantly faster while outperforming Dis-PFL. Thanks to its efficient Dynamic Split Scheduling (DSS) scheme, `MSfusion` achieves competitive performance in less than one-third of the computation time. A comprehensive overview of local efficiency, specifically for CIFAR-100, is provided in Table. 3. Notably, owing to the application of dynamic overlap in `MSfusion`, there exists a slight discrepancy in communication costs between the initiation and termination of training. Both values are provided, with the communication cost during training falling within this range. These findings underscore that `MSfusion` entails only 1/20 to 1/15 of the communication cost and 1/20 of the computation cost, a testament to its efficient overlap averaging methodology.

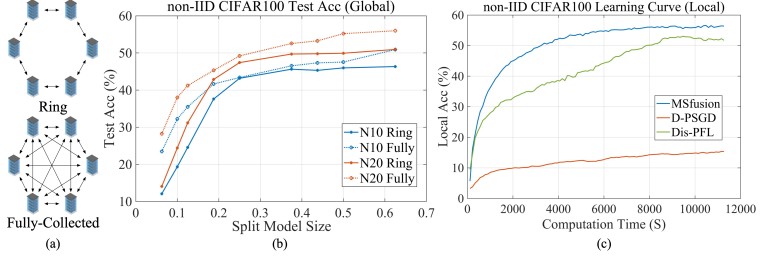

Figure 5: (a) Illustrations of ring and fully-Collected Nework topology. (b) Performance difference for `MSfusion` under ring and fully-Collected ($\mu = 50\%$ for all participants). (c)Performance comparison under ring topology (`MSfusion` $\mu = 25\%$ for all participants).

## 6 CONCLUSION

In this paper, we have introduced and evaluated `MSfusion`, a solution designed to address challenges in collaboratively training large models. Through its decentralized approach and an ensemble of techniques like the DSS scheme and overlap aggregation method, `MSfusion` strikes a balance between performance and efficiency. Its adaptive overlap strategy coupled with a tailored contrastive objective further sets it apart by ensuring convergence speed and mitigating model drift, respectively. Our empirical results underscore the potential of `MSfusion` to collaboratively training large models, offering a good way for organizations striving to maximize performance while judiciously managing resources.

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

# A    APPENDIX

## A.1    MATHEMATICAL REASONING FOR DSS

In this paper we mainly analysis the intution mathematical reasoning behind our proposed `MSfusion`.

For the collaborative learning problem considered in the (2) with $N$ participants and each with its own dataset $\mathcal{D}_n = \{(x_i^{(n)}, y_i^{(n)})\}_{i=1}^{M_n} \in \mathbb{R}^d \times \mathbb{R}$ . It can be summarized as the following problem:

$$
\begin{aligned}
\min_W \ &f(W) = \frac{1}{N} \sum_{n=1}^{N} f_n(w_n) \\
\text{s.t. } \ &f_n(w_n) := \frac{1}{K} \sum_{i=1}^{K} \mathcal{L}_n(w_n; (x_i, y_i)) \\
&w_n \subseteq W
\end{aligned}
\tag{9}
$$

In order to better investigating the DSS scheme, the definition of unbiased compressor is introduced following Beznosikov et al. (2020); Shulgin & Richtárik (2023).

**Definition 1.** Let $\zeta \geq 1$ and $\forall x \in \mathbb{R}^d$, for a (possibly random) mapping $\mathcal{C} : \mathbb{R}^d \to \mathbb{R}^d$, we say $\mathcal{C}$ is unbiased compressor operators ($\mathcal{C} \in \mathbb{U}(\zeta)$) if the following holds:

$$
\mathbb{E}[\mathcal{C}(x)] = x, \quad \mathbb{E}\left[\|\mathcal{C}(x)\|^2\right] \leq \zeta \|x\|^2
\tag{10}
$$

Like for a random splitting scheme splitting $q \in [d] := \{1, ..., d\}$ splitting the full model $W$, it can be viewed as a operator achieving the following

$$
\mathcal{C}_{\text{Random}}(W) := \mathbf{C}_{\text{Rand}} W = \frac{d}{q} \sum_{i \in Q} e_i e_i^\top W
\tag{11}
$$

where in random splitting scheme $Q \subseteq [d]$ is $q$ random sampling (a subset of $[d]$ of cardininality $q$ random chosen uniformly), $e_1, ..., e_d$ are standard unit basis vectors, and $\mathcal{C}_{\text{Random}}(W)$ belongs to $\mathbb{U}(\frac{d}{q})$, for a smaller size of split model (lower $q$), the higher the variance $\zeta$ of the compressor.

The Stochastic Gradient Descent (SGD) for participant $n$ with local model $w_n^r$ in round $r$ can be written as:

$$
w_n^{r+1} := w_n^r - \eta \nabla \mathcal{L}_n(w_n^r)
\tag{12}
$$

where $\eta$ is the step size. In this paper, we consider the splitting scheme offers a sketch compressor $\mathbf{C}_n^r \in \mathbb{R}^d \times \mathbb{R}$ to achieve sketching on global full model $W$. And the split submodel computation can be represented as the following:

$$
W^{r+1} = \mathbf{C}_n^r W^r - \eta \mathbf{C}_n^r \nabla \mathcal{L}_n(\mathbf{C}_n^r W^r)
\tag{13}
$$

The sketch $\mathbf{C}_n^r$ requires to be symmetric positive semi-definite matrix. The ideal of (13) is to reduce the cost of directly computation on full model $W^r$ since the compressor $\mathbf{C}_n^r$ ensure the update lies in a lower dimensional subspace. Each participant only compute its own the split model $w_n^r = \mathbf{C}_n^r W^r$, and we need this split model can effectively representing the global full model.

We mainly discuss the case where the global full model size is smaller than split model times participants number ($1 \leq \mu N$). In this situation, the Inter-Participant gap in DSS scheme ensures that the global full model $W$ can be fully covered by participants in each round. That is in round $r$, we have inter-participant compressor

$$
\mathbf{C}_{\text{part}}^r := \frac{1}{n} \sum_{i=1}^{n} \mathbf{C}_i^r = \mathbf{I}
\tag{14}
$$

where $\mathbf{I}$ is the identity matrix. And this means that in each round viewed from all the participants, the (13) is equivalent to the (12).

For the inter-round gap in DSS scheme ensures that the parameters of the global model $W$ are uniformly optimized by individual participants. That is for participant $n$, there exist $r^*$ to let the inter-round compressor

$$\mathbf{C}^n_{\text{round}} := \frac{1}{q^*} \sum_{i=1}^{r^*} \mathbf{C}^i_n = \mathbf{I} \tag{15}$$

where $q^* \in [d]$ is determined by the split model rate $\mu$. And this means means that for each participants viewed from finite round $r^*$, the (13) is equivalent to the (12).

Since from the Definition. 1, $\mathbb{E}\left[\|\mathcal{C}(x)\|^2\right] \leq \zeta\|x\|^2$, the overlapping parts can be viewed as an other unbiased overlapping compressor $\mathbf{C}^n_{\text{over}}$:

$$\mathbf{C}^n_{\text{over}} := N \cdot \sum_{j \in S} e_{\pi_j} e_{\pi_j}^\top \tag{16}$$

where $\pi_j$ is subset of $[d]$ determined by the overlapping part between participant $j$ and participant $n$, and in `MSfusion` this can be controled by $c$ in $dy_c$. $S \subseteq\in N$ is a set of participants connected with $n$ and have overlapping part.

Then we will make some commonly used assumptions to facilitate the analysis.

**Assumption 1.** The local loss function $\mathcal{L}_n$ is $\mathbf{L}$-smooth and $u$ strongly convex. That is $\forall x, y \in \mathbb{R}^d$, there exist a positive semi-definite matrix $\exists \mathbf{L} \in \mathbb{R}^d \times \mathbb{R}^d$

$$\mathcal{L}_n(y) \leq \mathcal{L}_n(x) + \langle \nabla \mathcal{L}_n(x), y - x \rangle + \frac{\mathbf{L}}{2}\|y - x\|_2^2 \tag{17}$$

and a positive semi-definite matrix $\exists u \in \mathbb{R}^d \times \mathbb{R}^d$

$$\mathcal{L}_n(y) \geq \mathcal{L}_n(x) + \langle \nabla \mathcal{L}_n(x), y - x \rangle + \frac{u}{2}\|y - x\|_2^2 \tag{18}$$

**Assumption 2.** In each communication round $r$, all the participant can computes the true gradient $\mathbf{C}^r_n \nabla \mathcal{L}_n(\mathbf{C}^r_n W^r)$ through its local submodel $w^r_n = \mathbf{C}^r_n W^r$.

To better study properties for DSS, we simplify the problem (9) into a quadratic problem

$$f(W) = \frac{1}{N} \sum_{n=1}^N f_i(w_n), \quad f_n(w_n) \equiv \frac{1}{2} w_n^\top \mathbf{L}_n w_n - w_n^\top \mathbf{b}_i \tag{19}$$

And under this simplification, $f(x)$ is $\overline{\mathbf{L}}$-smooth, and $\nabla f = \overline{\mathbf{L}}x - \overline{b}$ with $\overline{\mathbf{L}} = \frac{1}{n} \sum_{n=1}^N \mathbf{L}_i$ and $\overline{b} = \frac{1}{n} \sum_{n=1}^N b_i$.

We mainly examine the case of $b_i \equiv 0$, in this situation, the overall updating can be written as:

$$\frac{1}{N} \sum_{n=1}^N \mathbf{C}^r_n \nabla f_i(\mathbf{C}^r_n w_n) = \frac{1}{N} \sum_{n=1}^N \mathbf{C}^r_n \mathbf{L}_i \mathbf{C}^r_n w_n = \overline{\mathbf{B}}^r w^r_n \tag{20}$$

Proved in Shulgin & Richtárik (2023), we have the following theorem.

**Theorem 1.** *Consider a distributed learning setting with learning process shown in (20) for a quadratic problem (19) with $\overline{\mathbf{L}} \succ 0$ and $b_i \equiv 0$. Then for $\overline{A} := \frac{1}{2}\mathbb{E}[\overline{\mathbf{L}}\overline{\mathbf{B}}^r + \overline{\mathbf{L}}^r\overline{\mathbf{B}}] \succ 0$, there exists a constant $\xi > 0$*

$$\mathbb{E}[\overline{\mathbf{B}}^r \overline{\mathbf{L}} \overline{\mathbf{B}}^r] \preceq \xi \overline{A} \tag{21}$$

*and for a step size $\eta(0 < \eta < \frac{1}{\xi})$ the iterates satisfy the following:*

$$\frac{1}{R} \sum_{r=0}^{R-1} \mathbb{E}\left[\|\nabla f(w^r_n)\|^2_{\overline{\mathbf{L}}^{-1}\overline{\mathbf{AL}}^{-1}}\right] \leq \frac{2\left(f(w^0_n) - \mathbb{E}\left[f(w^R_n)\right]\right)}{\eta R} \tag{22}$$

*and*

$$\mathbb{E}\left[\|w^r_n - w^*_n\|^2_{\overline{\mathbf{L}}}\right] \leq \left(1 - \eta\lambda_{\min}\left(\overline{\mathbf{L}}^{-\frac{1}{2}}\overline{\mathbf{AL}}^{-\frac{1}{2}}\right)\right)^k \|w^r_n - w^*_n\|^2_{\overline{\mathbf{L}}} \tag{23}$$

*where $\lambda_{\min}()$ denotes minimum eigenvalue, $w^*_n := \arg\min f(w_n)$.*

By applying Theorem 1, inter-participant compressor where $\mathbf{C}_{\text{part}}^r = \mathbf{I}$, we have $\overline{\mathbf{B}}^r = \overline{\mathbf{L}}$, $\overline{\mathbf{B}}^r \overline{\mathbf{L}} \overline{\mathbf{B}}^r = \overline{\mathbf{L}}^3$ and $\overline{\mathbf{A}} = \overline{\mathbf{L}}^2 \succ 0$. So the (21) is satisfied for constant $\xi = \lambda_{max}(\overline{\mathbf{L}})$. And with a step size $\eta = \frac{1}{\xi}$, from (22) we have

$$\frac{1}{R} \sum_{r=0}^{R-1} \|\nabla f(w_n^r)\|_{\mathbf{I}}^2 \leq \frac{2\lambda_{max}(\overline{\mathbf{L}})(f(w_n^0) - f(w_n^R))}{R} \tag{24}$$

Which means with inter-participant compressor viewed from all the participants the problem will converge. And the analysis is the same for inter-round compressor.

Then is the analysis for overlapping compressor $\mathbf{C}_{\text{over}}^n$. For the case each participant with same split model size $\mu$, we have $f_n(w_n^r) = \frac{1}{2} {w_n^r}^\top \mathbf{L} w_n^r$ with $\mathbf{L} \equiv \mathbf{L}_n$. If we define a diagonal matrix $\mathbf{D} = \text{diag}(\mathbf{L})$. And then 19 can be changed into

$$f_n(\mathbf{D}^{-\frac{1}{2}} w_n^r) = \frac{1}{2} (\mathbf{D}^{-\frac{1}{2}} w_n^r)^\top \mathbf{L} (\mathbf{D}^{-\frac{1}{2}} w_n^r) = \frac{1}{2} (w_n^r)^\top (\mathbf{D}^{-\frac{1}{2}} \mathbf{L} \mathbf{D}^{-\frac{1}{2}}) w_n^r = \frac{1}{2} (w_n^r)^\top \hat{\mathbf{L}} w_n^r \tag{25}$$

where $\hat{\mathbf{L}} \succ 0$ as $\mathbf{L} \succ 0$, and $\text{diag}(\hat{\mathbf{L}}) = \mathbf{I}$. Since for each participant the overlapping rate with other participant is the same. Here we mainly analysis the overlapping part between each two neighbor participant with biggest overlapping rate, then for overlapping compressor at each round $r$ we have $\mathbf{C}_n^r = N \cdot e_{\pi_n^r} e_{\pi_n^r}^\top$, where $\pi_n^r$ is the overlapping part between $n$ and its neighbor with biggest overlapping rate. So we have

$$\mathbb{E}\left[\overline{\mathbf{B}}^r\right] = \mathbb{E}\left[\frac{1}{N} \sum_{n=1}^{N} \mathbf{C}_n^r \hat{\mathbf{L}}_i \mathbf{C}_n^r\right] = N \cdot \text{diag}(\hat{\mathbf{L}}) = N\mathbf{I} \tag{26}$$

Then the 21 can be transformed as

$$\xi\hat{\mathbf{A}} = \xi N\mathbf{I} \succeq N^2\mathbf{I} \tag{27}$$

which holds with $\xi \geq N$. And the 22 can be converted to

$$\|\nabla f(w_n^r)\|_{\hat{\mathbf{L}}^{-1} \hat{\mathbf{A}} \hat{\mathbf{L}}^{-1}}^2 \geq N\lambda_{\min}\left(\hat{\mathbf{L}}^{-1}\right) \|\nabla f(w_n^r)\|_{\mathbf{I}}^2 = N\lambda_{\max}(\hat{\mathbf{L}}) \|\nabla f(w_n^r)\|_{\mathbf{I}}^2 \tag{28}$$

So the convergence guarantee for the overlapping compressor is

$$\frac{1}{R} \sum_{r=0}^{R-1} \|\nabla f(w_n^r)\|_{\mathbf{I}}^2 \leq \frac{2\lambda_{max}(\hat{\mathbf{L}})(f(w_n^0) - \mathbb{E}\left[f(w_n^R)\right])}{R} \tag{29}$$

## A.2 EXPERIMENT DETAILS AND MORE RESULTS

### A.2.1 DATASETS DETAILS

CIFAR10 and CIFAR100 datasets each comprise 60,000 32x32 color images distributed across 10 and 100 classes, respectively, with 50,000 images used for training and 10,000 for testing. TinyImageNet, a scaled-down counterpart of the renowned ImageNet dataset, encompasses 200 classes derived from 100,000 224x224 images.

### A.2.2 DESCRIPTIONS OF HYPERPARAMETERS AND PLATFORM

For all the experiment, SGD optimizer is applied. The communication round for CIFAR10 and CIFAR100 experiments is 500, for TinyImageNet, PennTreebank and WikiText2 experiments is 800, for WikiText103 experiments is 200. Local epoch for particitants is 1. $\tau = 0.5$ like in Chen et al. (2020). $\lambda = 1$ following Li et al. (2021). The initial conrol parameter $c_0 = 1$, and final stage parameter $p = 0.75$. Table. 4 present the parameter size, data partition, and model architecture for each experiment.

Learning rate scheduler for MSfusion is CyclicLR scheduler which varies the learning rate between the minimal and maximal thresholds. The learning rate values change in a cycle from more minor to higher and vice versa. The reasons for choosing CyclicLR is the dynamic mechanisms in MSfusion. The minimal thresholds is set 0.001, the maximal is set 0.0012, the cycle round is the same with the

maximum communication round. Note to ensure equitable comparisons, we maintain uniformity in parameters across all PT-based baselines(HeteroFL and FedRolex), including learning rate, and the number of communication rounds. Other hyperparameters of FedRolex and HeteroFL is set by following Alam et al. (2023).

All the experiments are conducted using PyTorch version 2.0 on a single machine equipped with AMD EPYC 7542 CPU, 384GB of memory, and four NVIDIA 4090 GPUs.

Table 4: Experiments details.

| | CIFAR10 | CIFAR100 | TinyImageNet | PennTreebank | WikiText2 | WikiText103 |
|---|---|---|---|---|---|---|
| Data/Token size | 50,000 | 50,000 | 100,000 | 929,500 | 2,088,600 | 103,227,000 |
| Local data/token size (10 participants) | 5,000 | 5,000 | 10,000 | 92,950 | 208,860 | 10,322,700 |
| Local epoch | 1 | 1 | 1 | 1 | 1 | 1 |
| Batch size | 10 | 10 | 40 | 100 | 100 | 300 |
| Model applied | ResNet18 | ResNet18 | ResNet18 | Transformer | Transformer | Transformer |
| Hidden size | [64, 128, 256, 512] | [64, 128, 256, 512] | [64, 128, 256, 512] | [512, 512, 512, 512] | [512, 512, 512, 512] | [256, 256, 256, 256] |
| Embedding Size | | N/A | | 256 | 256 | 256 |
| Number of heads | | N/A | | 8 | 8 | 8 |
| Dropout | | N/A | | 0.2 | 0.2 | 0.2 |
| Sequence length | | N/A | | 64 | 64 | 64 |
| Parameter size of global full model | 11.2M | 11.2M | 11.3M | 7.32M | 19.3M | 139.01M |

### A.2.3 EFFECT OF INITIAL DYNAMIC GAP CONTROL PARAMETER

We also report the effect of initial dynamic gap control parameter $c_0$, results are given in Figure. 6. It can be shown that there is a huge performance difference between without inter-participant gap ($c_0 = 0$) and with inter-participant gap. Then the performance gradually increase until a optimal $c_0^*$, and this shown there is a trade-off between overlapping rate and global full model coverage each round with all participants.

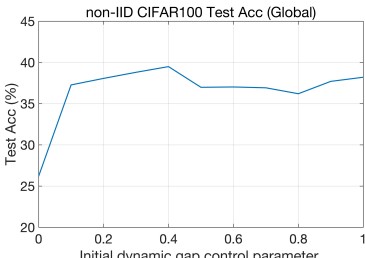

Figure 6: Effect of initial dynamic gap control parameter $c_0$

### A.2.4 HETEROGENEOUS SPLIT FUSION SETTING

We also discuss the heterogeneous split fusion setting where participants' server with different computation power are collaboratively training a large model. Specifically, in this paper, our heterogeneous setting is more constrained with split model size for all participants is less than half of the global full model ($\forall \mu_n \leq 0.5$). Figure. 7 shows the heterogeneous split model size comparison with ($\mu_n \in \{0.5, 0.25, 0.1875, 0.125, 0.0625\}$), Figure. 8 shows the heterogeneous split model size comparison with ($\mu_n \in \{0.25, 0.1875, 0.125, 0.0625, 0.03125\}$). In these experiments, total of 10 participants are involved in the collaboratively training process, each 2 are assigned with a fixed unique split model size from the list sets. It is clear that the proposed `MSfusion` way outperformed STOA PT-based methods in the more constrained heterogeneous collaboratively learning settings in both IID and non-IID data distributions with much faster converge speed and higher accuracy. FedRolex and outperform HeteroFL with its round-rolling scheme, but the performance of FedRolex is greatly dropped with smaller split model size in Figure. 8. While `MSfusion` can still maintain a good performance thanks to its much more efficient DSS scheme. It can also be observed that the model heterogeneous will greatly enhence the model drift between the participants resulting greatly influence the performance in the non-IID settings.

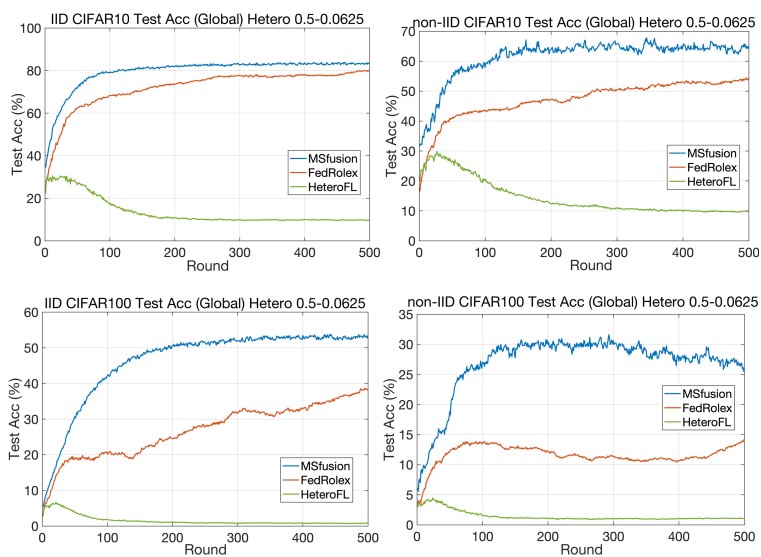

Figure 7: Heterogeneous split model size comparison ($\mu_n \in \{0.5, 0.25, 0.1875, 0.125, 0.0625\}$)

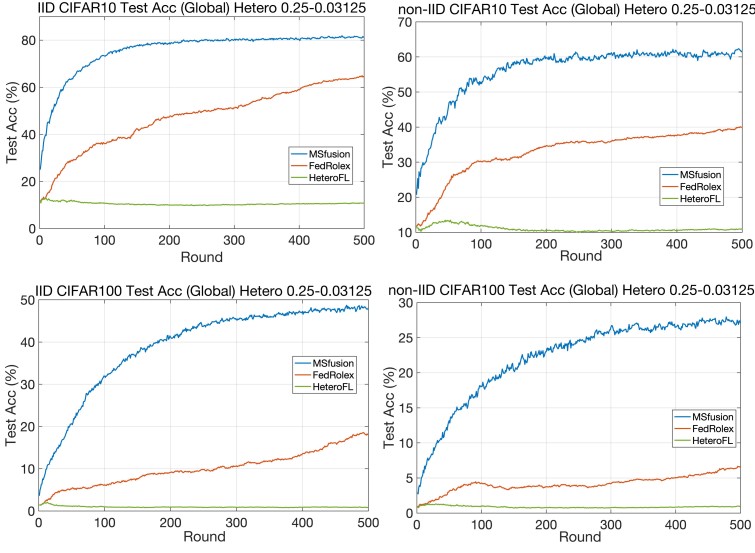

Figure 8: Heterogeneous split model size comparison ($\mu_n \in \{0.25, 0.1875, 0.125, 0.0625, 0.03125\}$)

