# OpenReview forum: "MSfusion: Enabling Collaborative Training of Large Models over Resource-Constraint Participants"
_ICLR.cc/2024/Conference — Submitted to ICLR 2024_

### Official Review · Reviewer_PcAY · 2023-10-29

**Soundness:** 3 good
**Presentation:** 3 good
**Contribution:** 3 good
**Rating:** 6
**Confidence:** 3

**Summary:**

This paper presents MSfusion, a collaborative learning framework that enables the training of large models on resource-constrained devices through model splitting. With its double shifting model splitting scheme, adaptive model overlapping, and contrastive loss functions, MSfusion maintains training effectiveness while significantly reducing computation and communication costs. The paper provides a mathematical reasoning for DSS and introduces the definition of an unbiased compressor. The authors also discuss the practical application of MSfusion in real-world scenarios where multiple companies with resource-limited servers and private data can collaborate to train high-performance large models. Overall, MSfusion offers a promising solution to the challenge of training large models on resource-constrained devices.

**Strengths:**

- The paper focuses on the significant task of collaborative training, which plays a vital role in various scenarios, particularly in training large models. The authors provide a clear motivation for their work, emphasizing the challenges faced when training resource-intensive models on devices with limited resources, and the necessity of collaborative learning frameworks. The practical application of MSfusion is also discussed, highlighting real-world scenarios where multiple companies with resource-limited servers and private data can collaborate to train high-performance large models.
- A highly effective collaborative training framework, MSfusion, is proposed in the paper, offering both computational efficiency and high performance. MSfusion leverages model splitting to enable effective and efficient training of large models across participants with resource constraints. The paper introduces the double shifting model splitting scheme, adaptive model overlapping, and contrastive loss functions to maintain training effectiveness while significantly reducing computation and communication costs. The authors provide a detailed description of the MSfusion framework, including the training process, communication protocol, and model aggregation method.
- The proposed approach is validated through experiments on various tasks. The authors conduct experiments on several datasets, such as CIFAR-10, CIFAR-100, and TinyImageNet, to evaluate the performance and efficiency of MSfusion. The results demonstrate that MSfusion achieves comparable or even superior performance compared to state-of-the-art methods, while substantially reducing computation and communication costs. The authors also perform ablation studies to analyze the contribution of each component of MSfusion to its overall performance.
- The paper includes in-depth theoretical analysis to support the proposed framework. This analysis encompasses the mathematical reasoning for the double shifting model splitting scheme and the definition of an unbiased compressor. The authors provide mathematical reasoning for DSS, which serves as a crucial component of MSfusion. Additionally, they introduce the definition of an unbiased compressor, which is employed to compress the model updates before transmission, thereby reducing communication costs. Theoretical analysis is provided to support the effectiveness of these components and their contribution to the overall performance of MSfusion.

**Weaknesses:**

- The performance gap between the ablated models and the proposed model is not very large. This does not support the importance of the proposed contrastive learning objective and the dynamic overlapping method.
- The association between the proposed collaborative training framework and the contrastive learning objective could be further discussed.  This additional loss seems independent from the proposed collaborative training method. Also, this contrastive learning method itself may benefit other distributed learning frameworks, or the backbone model itself.

**Questions:**

I would expect more discussion on the results of the ablation study, and the proposed cross-sub-model contrastive learning, as specified in the weakness part.

---

> ### Author Response · Authors · 2023-11-18
>
> Thank you for your thoughtful and positive feedback. Here, we provide answers to your concerns point by point.
>
> **Weaknesses 1:** Concern about the importance of the proposed contrastive learning objective and the dynamic overlapping method.
>
> **Our response:** Thank you for your insightful assessment. The crux of our incorporation of the contrastive learning objective and dynamic overlapping method within MSfusion aimed to address the divergence between local participant models and the aggregated model, aiming for notable performance enhancements. To better substantiate the importance of these adaptations, we've extended our experiments to include the WikiText2 dataset, focusing on ablation studies. The results are shown in the following Table which is updated to Table 2 in the revision Section 5.1.
>
> | Methods                | CIFAR10 ACC             | FLOPs       | WikiText2 PPL           | FLOPs       |
> | :--------------------- | :---------------------: | :---------: | :---------------------: | :---------: |
> | HeteroFL               | 37\.36 $\pm$ 0.6     | 35\.76M     | 579\.05 $\pm$ 8.4    | 1\.08B      |
> | FedRolex               | 66\.41 $\pm$ 0.8     | 35\.76M     | 547\.32 $\pm$ 4.5    | 1\.08B      |
> | MSfusion w/o Con       | 73\.16 $\pm$ 0.5     | 20\.32M     | 9\.57 $\pm$ 2.1      | 267\.2M     |
> | MSfusion w/o Dyn       | 73\.31 $\pm$ 0.5     | 22\.33M     | 7\.35 $\pm$ 1.5      | 290\.1M     |
> | MSfusion w/o Con & Dyn | 70\.17 $\pm$ 0.8     | **20\.32M** | 11\.24 $\pm$ 2.3     | **267\.2M** |
> | MSfusion               | **75\.71 $\pm$ 0.5** | 22\.33M     | **5\.276 $\pm$ 0.4** | 290\.1M     |
>
> Notably, with a split model size of $\mu=18.75\% $ in the WikiText2 dataset experiments, the performance gain of MSfusion over the baseline variants are more significent than the CV. This main attributed to the larger relative size of local model parameters within the transformer architecture, where the model drift problem is further amplified.
>
> These additional experiments underscore the tangible benefits and substantial impact of integrating these components into MSfusion, further solidifying their role in enhancing overall performance. We believe these supplementary findings reinforce the significance of the contrastive learning objective and dynamic overlapping method within MSfusion, lending stronger support to their effectiveness in improving performance.
>
> **Weaknesses 2:** Need clarification for the association between the proposed collaborative training framework and the contrastive learning objective.
>
> **Our response:** In our proposed decentralized collaborative training algorithm, we've highlighted two pivotal components: model shifting on a single server and model aggregation across servers. These aspects inherently introduce divergence within the local split model across rounds, as well as divergence between the local split model and the global full model. To explicitly address these divergence issues, our design integrates a contrastive learning loss that leverages representations from the local training model, local aggregated model, and the local model from the preceding round. This strategic utilization of contrastive learning aims to mitigate the challenges associated with model shifting within our MSfusion method. It's important to note that this contrastive learning objective is specifically tailored for our sub-model shifting algorithm, uniquely crafted for decentralized sub-model learning. This customized approach is a distinct facet of our decentralized framework, providing a targeted solution to the challenges of divergence encountered in collaborative training scenarios.
>
> This contrastive learning method, while intricately woven into our collaborative framework, also holds potential applicability beyond our specific framework. Its adaptability to other distributed learning frameworks or its potential benefits to the backbone model itself can indeed be an area of further exploration. However, the unique synergy between the contrastive learning objective and our decentralized sub-model shifting algorithm remains a central and distinctive aspect of our proposed MSfusion method.

---

> ### Author Response · Authors · 2023-11-18
>
> **Question:** Expectation for more discussion on the results of the ablation study.
>
> **Our response:** Your feedback on the need for a more comprehensive discussion regarding the results of the ablation study and the proposed cross-sub-model contrastive learning is greatly appreciated. In response, we've taken careful consideration of these aspects and included an expanded discussion in our revised manuscript. Additional experiments are included in Table 2 and the following discussion on the result is added in Section 5.1 to the revision.
>
> **The results showcase MSfusion's superior performance over all baseline variants. Specifically, in NLP tasks, the performance gain of MSfusion is more significant. This is mainly attributed to the larger relative size of local model parameters within the transformer architecture, further amplifying the model drift problem.**

---

> ### Author Response · Authors · 2023-11-20
>
> Dear Reviewer PcAY,
>
> We are approaching the end of the discussion phase, and we have unfortunately received no feedback from you on our rebuttal.
>
> Please can we kindly ask you to take a look at our responses, and let us know whether we have clarified your questions and addressed your concerns?
>
> Specifically, we've added new experiments for the ablation studies to better show the performance gain of MSfusion. We have clarified the association between the proposed collaborative training framework and the contrastive learning objective. And added more discussion about the ablation studies to the revision.
>
> Thank you very much again for the time you spent reviewing.
>
> Paper3368 Authors

---

> ### Author Response · Authors · 2023-11-22
> **A gentle reminder for Reviewer PcAY**
>
> Dear Reviewer PcAY,
>
> With less than 24 hours remaining in the discussion phase, we regret to note that we haven't received any feedback from you regarding our rebuttal.
>
> Please can we kindly ask you to take a look at our responses, and let us know whether we have clarified your questions and addressed your concerns?
>
> Specifically, we've added new experiments for the ablation studies to better show the performance gain of MSfusion. We have clarified the association between the proposed collaborative training framework and the contrastive learning objective. And added more discussion about the ablation studies to the revision.
>
> Thank you very much again for the time you spent reviewing.
>
> Paper3368 Authors

---

> ### Author Response · Authors · 2023-11-23
> **A gentle reminder for Reviewer PcAY**
>
> Dear Reviewer PcAY,
>
> With less than 3 hours remaining in the discussion phase, we regret to note that we haven't received any feedback from you regarding our rebuttal.
>
> Please can we kindly ask you to take a look at our responses, and let us know whether we have clarified your questions and addressed your concerns?
>
> Specifically, we've added new experiments for the ablation studies to better show the performance gain of MSfusion. We have clarified the association between the proposed collaborative training framework and the contrastive learning objective. And added more discussion about the ablation studies to the revision.
>
> Thank you very much again for the time you spent reviewing.
>
> Paper3368 Authors

---

### Official Review · Reviewer_wET1 · 2023-10-31

**Soundness:** 2 fair
**Presentation:** 3 good
**Contribution:** 2 fair
**Rating:** 3
**Confidence:** 4

**Summary:**

This work aims at a collaborative training framework of large models for resource-constrained participants. To solve this problem, model splitting and contrastive loss functions are adopted. The research problem is interesting and important, and technical solution is reasonable and easy to follow. However, there is a gap between the research background and the experimental setup. The most serious issue is that the models used in the experiments of the paper cannot be called as large models. Besides, both of the two adopted techniques, i.e., model splitting and contrastive loss have been widely explored in many existing federated learning approaches, leading to novelty concerns.

**Strengths:**

1.	The research problem mentioned in Introduction is interesting and important.
2.	The overall structure and writing of the paper are well-organized and clear, enhancing the readability and understanding of the content.
3.	The adopted techniques are reasonable.

**Weaknesses:**

1.	Lacks of novel contribution. The proposed framework leverages adaptive model overlapping and contrastive loss function. These two proposed techniques, i.e., model splitting and contrastive loss have been investigated by many existing researches, while I do not find sufficient discussion on the related works. To have a clear view on the contribution, it is reasonable to have a comparison between the proposed approach and [1][2].
2.	There is a gap between the experimental setup and the research background of the paper. From the title and abstract of the paper, I was looking forward to seeing collaborative training of large models, especially since the first sentence of the abstract mentioned models like GPT-3. However, I was surprised to find in the experimental setup that the visual tasks were performed using ResNet-18, and the model used for text-related tasks was not explicitly mentioned. But based on the calculated FLOPS in Table 1, I can infer that the parameter size of this model is far less than 1B. I am curious if a model with such parameter size can be considered a large model. If the authors decide to claim this framework is designed for Large Model, I highly recommend to conduct experiments on real large models such as models with at least billion-sized parameters.
3.	Insufficient baselines. The authors claim that the proposed framework is compared with partial training-based approaches, however, there are some well-recognized approaches missed, e.g., [1][3].
4.	Lacks of theoretical analysis. Considering the there is a proposed loss function in the framework, it is better to have a theoretical convergence analysis on the framework.
5.	There are some typos, e.g., " for TinyImageNet experiments for WikiText2 experiments is 800" in section A.2.2.


Mentioned Reference

[1] Thapa, Chandra, et al. "Splitfed: When federated learning meets split learning." Proceedings of the AAAI Conference on Artificial Intelligence. Vol. 36. No. 8. 2022.

[2] Li, Qinbin, Bingsheng He, and Dawn Song. "Model-contrastive federated learning." Proceedings of the IEEE/CVF conference on computer vision and pattern recognition. 2021.

[3] Collins, Liam, et al. "Exploiting shared representations for personalized federated learning." International conference on machine learning. PMLR, 2021.

**Questions:**

Please see the weakness above.

**Details Of Ethics Concerns:**

None.

---

> ### Author Response · Authors · 2023-11-18
>
> Thank you for your review and thoughtful comments and suggestions. We have updated our paper based on your feedback and we will also address each point raised in the following separate comments.
>
> **Weaknesses 1:** Novelty concern toward the previous splitting learning and FL works.
>
> **Our response:** Please note while we use the word "model splitting" in our approach, there is fundamental difference between model splitting approach in this paper and the "split learning" in Splitfed framework [1]. In our decentralized collaborative learning setting, there is no centralized server, and the model are split across different participants. In Splitfed, or more generally split learning, the model is split across a central server and clients, who work together to train the entire model through communicating intermediate data embeddings and gradients. Due to splitting of the model, the clients can not train its own part independently due to missing computations of the submodel at the server side.  In sharp contrast, for the setting considered in this paper and the proposed MSfusion scheme, each participant is assigned a (dynamically changing) subset of model parameters, and **the training of these parameters are done independently at each participant**.  We believe that this difference can be well illustrated by Figure 1 and Figure 2 in our paper, and Figure 1 in Splitfed paper.
>
> While contrastive learning was utilized in Moon [2] to deal with data heterogeneity in federated learning, we use constrastive learning with a drastically distinct purpose of addressing the divergence within the local split model across rounds, as well as divergence between the local split model and the global full model, arising from the double shifting model splitting scheme introduced by MSfusion.  To explicitly address these divergence issues, we design a novel contrastive learning loss in (7) that incorporates representations from the local training model, local aggregated model, and the local model from the preceding round. It's important to note that this contrastive learning objective is specifically tailored for our sub-model shifting algorithm, uniquely crafted for decentralized sub-model learning. This customized approach is a distinct facet of our decentralized framework, providing a targeted solution to the challenges of divergence encountered in collaborative training scenarios.
>
> To summarize, our designs of model splitting and contrastive loss have fundamental differences with Splitfed [1] and Moon [2] respectively, in terms of design goals and specific implementations.
>
> [1] Thapa, Chandra, et al. "Splitfed: When federated learning meets split learning." Proceedings of the AAAI Conference on Artificial Intelligence. Vol. 36. No. 8. 2022.
>
> [2] Li, Qinbin, Bingsheng He, and Dawn Song. "Model-contrastive federated learning." Proceedings of the IEEE/CVF conference on computer vision and pattern recognition. 2021.

---

> > ### Author Response · Authors · 2023-11-18
> >
> > **Weaknesses 2:** Concern about limited model size and claim for the large model.
> >
> > **Our response:** Thank you for pointing out this. Firstly, we note that many previous works claim to work for "large model", while the parameter sizes in their experiments do not reach billion level. In a well cited paper [4], experiments are conducted on 100 million size (GPT-2 117M); In an ICLR 2023 paper [5], they claim to work on "Large-scale linear models", while experiments are done on ResNet-18.
> >
> > Nevertheless, in order to better evaluate our proposed MSfusion for LLMs, we have now added two new wildly adapted NLP datasets PennTreebank and WikiText103. For PennTreebank, it contains about a million tokens with a vocab size of 10,000, and for WikiText103, it contains about 103 million tokens with a vocab size of 267,735. The global full model parameter size for transformers applied to these datasets is 7.32M for PennTreebank, 19.3M for WikiText2 and 139.01M for WikiText103. The results are shown in the following table, which we have added to the Table 1 in revision.
> >
> > |            | PennTreebank           |            |             | WikiText2              |             |             | WikiText103            |            |             |
> > | :--------- | :--------------------: | :--------: | :---------: | :--------------------: | :---------: | :---------: | :--------------------: | :--------: | :---------: |
> > | Methods    | Perplexity             | FLOPs      | $ \mu_n\$ | Perplexity             | FLOPs       | $ \mu_n\$ | Perplexity             | FLOPs      | $ \mu_n\$  |
> > | HeteroFL   | 55\.97 $\pm$  5.4    | 148\.6M    | 75%         | 579\.05 $\pm$  8.4   | 1\.08B      | 75%         | 784\.21 $\pm$  23    | 111\.9B    | 75%         |
> > | FedRolex   | 61\.52 $\pm$  6.8    | 148\.6M    | 75%         | 547\.32 $\pm$  45    | 1\.08B      | 75%         | 697\.42 $\pm$  33    | 111\.9B    | 75%         |
> > | Fed-ET     | N/A                    |            |             | N/A                    |             |             | N/A                    |            |             |
> > | MSfusion S | 9\.09 $\pm$  0.7     | **36\.2M** | 21\.875%    | 44\.33 $\pm$  3.6    | **198\.5M** | 18\.75%     |  13\.21 $\pm$ 1.1   | **32\.6B** | 21\.875%    |
> > | MSfusion M | 8\.02 $\pm$ 0.5     | 41\.8M     | 25%         | 5\.28 $\pm$  0.4     | 290\.1M     | 25%         | 8\.92 $\pm$  0.6     | 37\.3B     | 25%         |
> > | MSfusion L | **3\.11 $\pm$  0.2** | 91\.4M     | 50%         | **3\.59 $\pm$  0.2** | 633\.2M     | 50%         | **6\.31 $\pm$  0.4** | 74\.4B     | 50%         |
> >
> > We note that these are the largest models we can experiment on given our computing resources. In addition to WikiText2, now the evaluations of MSfusion include 3 computer vision tasks and 3 NLP tasks. From the updated Table 1, it shows that MSfuion achieves high performance with low computation cost.
> >
> > From these additional experiment results, it is reasonable for us to infer that our propose method is suitable for even larger models.
> >
> > [4] Tian, Yuanyishu, et al. "FedBERT: When federated learning meets pre-training." ACM Transactions on Intelligent Systems and Technology (TIST) 13.4 (2022): 1-26.
> >
> > [5] Antorán, Javier, et al. "Sampling-based inference for large linear models, with application to linearised Laplace." ICLR 2023
> >
> > **Weaknesses 3:** Insufficient baselines.
> >
> > **Our response:** The main reason why we have small number of baselines is that the problem setting in our paper is rarely investigated in the literature of FL. Hence there are limited number of methods we can fairly compare with.
> >    As detailed in the response to W1, there are fundamental differences between the settings and approaches between the listed papers [1] [2] and our proposed MSfusion. For FedRep in [3], we note that it cannot be considered as partial training-based method, since in FedRep the local training is still on full model parameters; and the model is only partially transmitted and aggregated at the central server to deal with the data heterogeneity problem.
> >
> > [3] Collins, Liam, et al. "Exploiting shared representations for personalized federated learning." International conference on machine learning. PMLR, 2021.
> >
> > **Weaknesses 4:** Lacks of theoretical analysis.
> >
> > **Our response:** We would like to remind the reviewer that we indeed had a convergence analysis of our proposed method, which was placed in Appendix A.1 due to page limit. We now add the following bold **Detail analysis for DSS are provided in Appendix A.1.** at Section 4.1 in our revision for easier identification. Our convergence analysis can also be viewed as an improvement over existing partial training works since no convergence analysis is provided in previous works like HeteroFL and FedRolex.
> >
> > **Weaknesses 5:** There are some typos.
> >
> > **Our response:** Thanks for your careful reading. We have now proof read the draft and corrected the typos.

---

> ### Author Response · Authors · 2023-11-20
>
> Dear Reviewer wET1,
>
> We are approaching the end of the discussion phase, and we have unfortunately received no feedback from you on our rebuttal.
>
> Please can we kindly ask you to take a look at our responses, and let us know whether we have clarified your questions and addressed your concerns?
>
> Specifically, we've clarified the distinctiveness of MSfusion in comparison to splitting methods and prior FL works, expanded our experiments by integrating larger transformer models for a more comprehensive evaluation, explained reasons why baselines are limited, emphasized our theoretical analysis in the paper and corrected typos in the revised version based on your suggestions.
>
> Thank you very much again for the time you spent reviewing.
>
> Paper3368 Authors

---

> ### Author Response · Authors · 2023-11-22
> **A gentle reminder for Reviewer wET1**
>
> Dear Reviewer wET1,
>
> With less than 24 hours remaining in the discussion phase, we regret to note that we haven't received any feedback from you regarding our rebuttal.
>
> Please can we kindly ask you to take a look at our responses, and let us know whether we have clarified your questions and addressed your concerns?
>
> Specifically, we've clarified the distinctiveness of MSfusion in comparison to splitting methods and prior FL works, expanded our experiments by integrating larger transformer models for a more comprehensive evaluation, explained reasons why baselines are limited, emphasized our theoretical analysis in the paper and corrected typos in the revised version based on your suggestions.
>
> Thank you very much again for the time you spent reviewing.
>
> Paper3368 Authors

---

> ### Author Response · Authors · 2023-11-23
> **A gentle reminder for Reviewer wET1**
>
> Dear Reviewer wET1,
>
> With less than 3 hours remaining in the discussion phase, we regret to note that we haven't received any feedback from you regarding our rebuttal.
>
> Please can we kindly ask you to take a look at our responses, and let us know whether we have clarified your questions and addressed your concerns?
>
> Specifically, we've clarified the distinctiveness of MSfusion in comparison to splitting methods and prior FL works, expanded our experiments by integrating larger transformer models for a more comprehensive evaluation, explained reasons why baselines are limited, emphasized our theoretical analysis in the paper and corrected typos in the revised version based on your suggestions.
>
> Thank you very much again for the time you spent reviewing.
>
> Paper3368 Authors

---

### Official Review · Reviewer_wTYd · 2023-11-04

**Soundness:** 2 fair
**Presentation:** 3 good
**Contribution:** 2 fair
**Rating:** 5
**Confidence:** 3

**Summary:**

Summary*
This work focuses on collaborative learning of large models over resource-constrained
participants. The authors propose a new model splitting strategy that assigns a submodel of the
full global model to each participant. They further introduce adaptive model overlapping and
contrastive loss functions, achieving effective and efficient training. The evaluation is conducted
over 3 image datasets (i.e., CIFAR10, CIFAR 100, and TinyImageNet) and 1 natural language
dataset (i.e., WikiText2) using ResNet18 and Transformer-based network.

**Strengths:**

1. The submodel splitting strategy is a natural way to reduce the computation cost and has
been well studied in the federated learning context. The key contribution of this work is the
design of adaptive model overlapping and contrastive loss functions to help maintain
training effectiveness against model shift across participants.
2. The convergence of the proposed algorithm is analyzed in smooth and strongly convex case.

**Weaknesses:**

1. The technical contribution of this paper is limited. Except introducing a double shifting
model splitting scheme, the proposed design has no significant difference from existing
partial training work as reviewed in Section 2.3.
2. The convex assumption over the loss function (i.e., Assumption 1) for convergence analysis is
quite strong.
3. The experiments cannot support the previous design and analysis sections. While the
previous design sections are claimed to study the collaborative training of large language
models (e.g., GPT-3 in Abstract), the evaluation diverges to computer vision tasks and only
take one natural language dataset for evaluation.
4. Some important evaluation details are missing, including the detailed parameter size and the
number of layers of the transformer model, as well as the size of participants and the
dataset partition for the experiment over WikiText2.

**Questions:**

From Table 1, why the baselines of HeteroFL and FedRolex perform so badly over WikiText2?

---

> ### Author Response · Authors · 2023-11-18
>
> Thank you for your thorough feedback. We offer the following responses to address your concerns and questions:
>
> **Weaknesses 1:** Contribution concern over existing partial training work.
>
> **Our response:** We would like to emphasize that the introduced double shifting model splitting scheme (DSS) requires careful design of splitting scheme across participants and aggregation rounds (including initial placement of submodels, and the dynamic adjustment of model overlaps). Together with another important novel design of contrastive loss function, the proposed DSS enables each participant to operate with equally minimum computation and communication loads, achieving good performance on large models that cannot be trained in entirety on a single participant.
>
> Note that the related works in Section 2.3, while also utilizing partial training, exhibit quite distinct goals and designs. For instance in [1], the goal of model splitting is to accommodate client heterogeneity, and achieving good model performance relies on existence of powerful clients who train the entire model locally. Such clients do not exist in our considered setting (all participants are resource constraint who cannot train the entire model locally), and designing model splitting scheme poses major challenges that are addressed in this paper.
>
> [1] Samiul Alam, Luyang Liu, Ming Yan, and Mi Zhang. FedRolex: Model-heterogeneous federated learning with rolling sub-model extraction, NeurIPS 2022.
>
> **Weaknesses 2:** The convex assumption over the loss function (i.e., Assumption 1) for convergence analysis is quite strong.
>
> **Our response:** We agree that the smoothness and strong convexity assumptions over the loss function are strong, but it is worth noting they are also widely adopted in the literature of distributed learning frameworks, like [2][3][4]. In these works, smoothness and strong convexity assumptions are also made to facilitate the convergence proofs. Our convergence analysis can also be viewed as an improvement over existing partial training works since no convergence analysis is provided in previous works like HeteroFL and FedRolex.
>
> [2] Aleksandr Beznosikov, et al. "On Biased Compression for Distributed Learning." arXiv:2002.12410, 2020.
>
> [3] Tian Li, et al. "Ditto: Fair and Robust Federated Learning Through Personalization" ICML 2021
>
> [4] Yifan Shi, et al. "Improving the Model Consistency of Decentralized Federated Learning" ICML 2023
>
> **Weaknesses 3:** Limited NLP evaluation.
>
> **Our response:** Thank you so much for pointing out this. In order to better evaluation our proposed MSfusion for LLMs, we have added two new wildly adapted NLP datasets PennTreebank and WikiText103. For PennTreebank, it contains about a million tokens with a vocab size of 10,000, and for WikiText103, it contains about 103 million tokens with a vocab size of 267,735. Together with WikiText2, our evaluation now covers 3 computer vision tasks and 3 NLP tasks, with different global full model parameter sizes, ranging from 7.32M for PennTreebank to 139.01M for WikiText103. The results are shown in the following table, which we have added to the Table 1 in the revised draft. The table demonstrates that our proposed MSfusion can successfully finish the collaborative training process with good performance, while greatly reducing the computation cost for each participant. We hope that this additional experiment can address the reviewer’s concern and strengthen our contribution.
>
> |            | PennTreebank           |            |             | WikiText2              |             |             | WikiText103            |            |             |
> | :--------- | :--------------------: | :--------: | :---------: | :--------------------: | :---------: | :---------: | :--------------------: | :--------: | :---------: |
> | Methods    | Perplexity             | FLOPs      | $ \mu_n\$ | Perplexity             | FLOPs       | $ \mu_n\$ | Perplexity             | FLOPs      | $ \mu_n\$  |
> | HeteroFL   | 55\.97 $\pm$  5.4    | 148\.6M    | 75%         | 579\.05 $\pm$  8.4   | 1\.08B      | 75%         | 784\.21 $\pm$  23    | 111\.9B    | 75%         |
> | FedRolex   | 61\.52 $\pm$  6.8    | 148\.6M    | 75%         | 547\.32 $\pm$  45    | 1\.08B      | 75%         | 697\.42 $\pm$  33    | 111\.9B    | 75%         |
> | Fed-ET     | N/A                    |            |             | N/A                    |             |             | N/A                    |            |             |
> | MSfusion S | 9\.09 $\pm$  0.7     | **36\.2M** | 21\.875%    | 44\.33 $\pm$  3.6    | **198\.5M** | 18\.75%     |  13\.21 $\pm$ 1.1   | **32\.6B** | 21\.875%    |
> | MSfusion M | 8\.02 $\pm$ 0.5     | 41\.8M     | 25%         | 5\.28 $\pm$  0.4     | 290\.1M     | 25%         | 8\.92 $\pm$  0.6     | 37\.3B     | 25%         |
> | MSfusion L | **3\.11 $\pm$  0.2** | 91\.4M     | 50%         | **3\.59 $\pm$  0.2** | 633\.2M     | 50%         | **6\.31 $\pm$  0.4** | 74\.4B     | 50%         |

---

> ### Author Response · Authors · 2023-11-18
>
> **Weaknesses 4:** Experiment details not clear.
>
> **Our response:** We agree that our previous evaluation details were not clear enough, and we have added a Table 4 to the revision in Appendix A.2.2 to present the parameter size, data partition, and model architecture for each experiment. The participant size for each experiment is emphasized in the paper. We hope that this table can provide a clearer understanding of our experimental settings and results.
>
> |                                               | CIFAR10             | CIFAR100            | TinyImageNet        | PennTreebank         | WikiText2            | WikiText103          |
> | :-------------------------------------------- | :-----------------: | :-----------------: | :-----------------: | :------------------: | :------------------: | :------------------: |
> | Data/Token size                               | 50,000              | 50,000              | 100,000             | 929,500              | 2,088,600            | 103,227,000          |
> | Local data/token size  (10 participants) | 5,000               | 5,000               | 10,000              | 92,950               | 208,860              | 10,322,700           |
> | Local epoch                                   | 1                   | 1                   | 1                   | 1                    | 1                    | 1                    |
> | Batch size                                    | 10                  | 10                  | 40                  | 100                  | 100                  | 300                  |
> | Model applied                                 | ResNet18            | ResNet18            | ResNet18            | Transformer          | Transformer          | Transformer          |
> | Hidden size                                   | [64, 128, 256, 512] | [64, 128, 256, 512] | [64, 128, 256, 512] | [512, 512, 512, 512] | [512, 512, 512, 512] | [256, 256, 256, 256] |
> | Embedding Size                                | N/A                 |                     |                     | 256                  | 256                  | 256                  |
> | Number of heads                               | N/A                 |                     |                     | 8                    | 8                    | 8                    |
> | Dropout                                       | N/A                 |                     |                     | 0\.2                 | 0\.2                 | 0\.2                 |
> | Sequence length                               | N/A                 |                     |                     | 64                   | 64                   | 64                   |
> | Parameter size of   global full model    | 11\.2M              | 11\.2M              | 11\.3M              | 7\.32M               | 19\.3M               | 139\.01M             |
>
> **Question:** From Table 1, why the baselines of HeteroFL and FedRolex perform so badly over WikiText2?
>
> **Our response:** We thank the reviewer for the question. The poor performance  in Table 1 regarding the baselines of HeteroFL and FedRolex over the WikiText2 dataset primarily stems from their inherent inability to address the challenge of model drift. As clarified in our paper, our study focuses on collaborative training of large models, where participants possess comparable yet limited split model sizes. In the experiments outlined in Table 1, the split model sizes for all participants are fixed. Particularly concerning transformers, the local split model parameters are relatively larger compared to ResNet in computer vision tasks. This discrepancy results in a situation where HeteroFL leaves a significant proportion of global full model parameters untrained. Similarly, FedRolex exacerbates the model drift problem by not training a considerable number of global full model parameters between communication rounds. Given that these methodologies primarily concentrate on the splitting scheme and do not address the model drift issue, their performance is notably compromised when collaboratively training larger models. To better elucidate these observations, we have included the following explanation in revision Section 5.1.
>
> **In NLP tasks, MSfusion significantly outperforms HeteroFL and FedRolex. The suboptimal performance of HeteroFL and FedRolex in collaborative training larger models is due to untrained global model parameters and exacerbated model drift, issues unaddressed in their splitting-focused methodologies. This emphasizes the advantage of MSfusion in handling larger models more effectively.**

---

> ### Author Response · Authors · 2023-11-20
>
> Dear Reviewer wTYd,
>
> We are approaching the end of the discussion phase, and we have unfortunately received no feedback from you on our rebuttal.
>
> Please can we kindly ask you to take a look at our responses, and let us know whether we have clarified your questions and addressed your concerns?
>
> Specifically, we've clarified our novel contributions in problem setting, split scheme design, dynamic overlapping, and contrastive loss design compared to previous partial training works. Additionally, we've addressed the utilization of smoothness and strong convexity assumptions in distributed learning, expanded our NLP experiments for a more comprehensive evaluation, introduced an experiment details table as suggested, and provided an explanation for the performance drop of HeteroFL and FedRolex in Table 1 within the revised version.
>
> Thank you very much again for the time you spent reviewing.
>
> Paper3368 Authors

---

> ### Author Response · Authors · 2023-11-22
> **A gentle reminder for Reviewer wTYd**
>
> Dear Reviewer wTYd,
>
> With less than 24 hours remaining in the discussion phase, we regret to note that we haven't received any feedback from you regarding our rebuttal.
>
> Please can we kindly ask you to take a look at our responses, and let us know whether we have clarified your questions and addressed your concerns?
>
> Specifically, we've clarified our novel contributions in problem setting, split scheme design, dynamic overlapping, and contrastive loss design compared to previous partial training works. Additionally, we've addressed the utilization of smoothness and strong convexity assumptions in distributed learning, expanded our NLP experiments for a more comprehensive evaluation, introduced an experiment details table as suggested, and provided an explanation for the performance drop of HeteroFL and FedRolex in Table 1 within the revised version.
>
> Thank you very much again for the time you spent reviewing.
>
> Paper3368 Authors

---

> ### Author Response · Authors · 2023-11-23
> **A gentle reminder for Reviewer wTYd**
>
> Dear Reviewer wTYd,
>
> With less than 3 hours remaining in the discussion phase, we regret to note that we haven't received any feedback from you regarding our rebuttal.
>
> Please can we kindly ask you to take a look at our responses, and let us know whether we have clarified your questions and addressed your concerns?
>
> Specifically, we've clarified our novel contributions in problem setting, split scheme design, dynamic overlapping, and contrastive loss design compared to previous partial training works. Additionally, we've addressed the utilization of smoothness and strong convexity assumptions in distributed learning, expanded our NLP experiments for a more comprehensive evaluation, introduced an experiment details table as suggested, and provided an explanation for the performance drop of HeteroFL and FedRolex in Table 1 within the revised version.
>
> Thank you very much again for the time you spent reviewing.
>
> Paper3368 Authors

---

### Author Response · Authors · 2023-11-18
**Global Response to Reviewers**

We are profoundly thankful to all reviewers for their insightful comments and critical examination of our work. Your collective insights have been instrumental in enhancing the quality and clarity of our work.

To address Reviewer wTYd’s concern about the lack of NLP task evaluation and wET1’s concern about the lack of larger model-size experiments. We have now added two new NLP datasets PennTreebank and WikiText103. Together with WikiText2, our evaluation now covers 3 computer vision tasks and 3 NLP tasks, with different global full model parameter sizes, ranging from 7.32M for PennTreebank to 139.01M for WikiText103. Results are shown in Table 1 at revision.

As suggested by Reviewer wTYd, we have added *table to present detailed settings of our experiments.

In response to Reviewer wTYd’s novelty concern toward the previous partial training works and Reviewer wET1’s novelty concern toward the previous splitting learning or FL works. We have further clarified technical contribution and novelty over these previous works on problem setting, split scheme design, dynamic overlapping, and customized contrastive loss design.

To address Reviewer PcAY’s expectation for more discussion about the ablation study, we added the ablation study in WikiText2 and explained the necessity and distinctiveness of contrastive loss objective in MSfusion.

We sincerely appreciate the time and effort put into the review process and look forward to any further comments or suggestions.

---

### Meta-Review · Area_Chair_kTfx · 2023-12-09

**Metareview:**

This work introduces MSfusion, a collaborative learning framework designed for training large models on resource-constrained devices. The authors propose a novel model splitting strategy, assigning submodels of the global model to participants, and incorporate adaptive model overlapping and contrastive loss functions to enhance training efficiency. MSfusion is evaluated on three image datasets (CIFAR10, CIFAR 100, and TinyImageNet) and one natural language dataset (WikiText2) using ResNet18 and Transformer-based networks. With a double shifting model splitting scheme, MSfusion effectively reduces computation and communication costs while maintaining training effectiveness. The paper provides mathematical reasoning for the double shifting scheme, introduces the concept of an unbiased compressor, and discusses practical applications in scenarios where multiple companies collaborate on training large models with resource-limited servers and private data.


While the motivation is strong, the reviewers find the contribution of the presented paper to be incremental. In particular, the primary ingredients of the proposal have been extensively studied in the literature. The above is perhaps okay if a strong engineering effort and results are presented, but the reviewers concern about the experiments are rather limited to “smaller” vision model and tasks, as compared to the promise of collaboratively training large models.

**Justification For Why Not Higher Score:**

The presented experiments do not seem to fully justify the motivation and claims made in this paper. The authors are encouraged to better position their contributions.

**Justification For Why Not Lower Score:**

N/A

---

### Decision · Program_Chairs · 2024-01-16

Reject